# Gender Differences in the Prevalence of and Factors Related to Non-Suicidal Self-Injury among Middle and High School Students in South Korea

**DOI:** 10.3390/ijerph18115965

**Published:** 2021-06-02

**Authors:** Jin-Young Jeong, Dong-Hyun Kim

**Affiliations:** 1Hallym Research Institute of Clinical Epidemiology, Hallym University, Chuncheon 24252, Korea; jjy@hallym.ac.kr; 2Department of Social and Preventive Medicine, Hallym University College of Medicine, Chuncheon 24252, Korea

**Keywords:** non-suicidal self-injury, adolescent, gender difference

## Abstract

This study was conducted to estimate the prevalence of non-suicidal self-injury (NSSI)—the most severe mental health problem among South Korean adolescents in 2019. The participants were students from 12 middle and seven high schools in a single city in the metropolitan area in South Korea. A cluster-random sampling method was used—one class per school year at each middle and high schools. Every student in the selected classes was investigated. The final number of selected classes and students were as follows: 36 classes, 1074 students at middle schools; 35 classes, 873 students at high school. Data were obtained using a self-administered online questionnaire. The 12-month prevalence of NSSI was 8.8% among Korean adolescents, 2.85 times higher for female than male students (13.4% vs. 4.7%). For the school grade, the multivariable-adjusted odds ratio [mOR] was 3.89 times (95% confidence interval [CI] 1.35–11.26) higher for the group with the highest school records compared with the middle group for male students, while such pattern was not observed among female students (mOR 1.11, 95% CI 0.54–2.26). The risks for NSSI were higher among female students with lower economic status, compared with those with higher economic status (mOR 4.16, 95% CI 1.43–12.05). In contrast, economic status did not indicate higher risk among male students (mOR 1.43, 95% CI 0.42–4.84). This study suggests that the risk for NSSI among adolescent students may be different according to gender and comprehensive interventions from schools against NSSI may need a gender-sensitive approach in South Korea.

## 1. Introduction

The leading cause of death among South Korean adolescents is suicide [1]. Looking at the trend of suicide rates among 10–19 year olds over the past two decades, it had peaked at 6.5 per 100,000 in 2009 and declined for a period, then started to rise again from 2016 (Figure 1). It is noticeable that the suicide rate is now rapidly increasing among teenage girls. The Organization for Economic Cooperation and Development (OECD) statistics also shows this problematic trend of soaring suicide rate among teenage girls in South Korea, where the rate of teenage girls was 6.4 per 100,000, 1.9 times higher than the OECD average (2.3 per 100,000 persons) in 2019. However, that of teenage boys is 5.5 per 100,000 in Korea, which is lower than the OECD average, 5.8 per 100,000 [2]. 

Suicidal behaviors (suicidal ideation, suicidal plans, and suicidal attempts) as well as suicide rates were high in female teenagers [3]. The rates of suicidal ideation, suicidal plans, and suicidal attempts among middle and high school female students in South Korea as of 2019 were 17.1%, 5.0%, and 4.0%, respectively, which are higher than those among male students with 9.4%, 3.1%, and 1.9%. The data clearly shows that Korean female teenagers are at higher risk of suicide.

The seriousness of teenagers’ self-injury has been only recently recognized. Concerns and interests are growing regarding adolescents’ self-injury as a series of media reports have been released that the number of online contents implying or demonstrating the attempts of self-injury has jumped [4] and that the number of counseling regarding self-injury has increased three times from the previous year [5]. Against this backdrop, the nation’s first three-month experience rate of self-injury in middle and high school students nationwide (first year middle school: 7.9%, first year high school: 6.4%) was published [6]. 

According to the DSM-5, the diagnostic and statistical manual of mental disorders [7], the criteria that conceptually differentiates NSSI from suicide attempts is the intention regarding death. The 12-month aggressive prevalence of NSSI produced by a meta-analysis among adolescents across the globe was 19.5% [8], which is 1.5 times higher than undergraduate students [9] and 3.5 times higher than adults [9]. Despite no intention to commit suicide, repeated attempts of NSSI for a long period time significantly increase subsequent suicidal ideations, suicidal attempts [10], and suicidal deaths [11].

Adolescence is a transitional period from children to adulthood in physical, psychological, and social aspects. Adolescents experience more stress, especially during this period, while trying to adapt to such changes [12]. According to the 2020 Review of Korean Children’s and Youth rights [13], among the cause of adolescents’ suicidal feelings, learning issues (academic pressure, school grades, etc.) were highest at 39.8%, followed by anxiety about future career (25.5%), family conflicts (16.0%), school violence (4.8%), and financial difficulties (1.7%). This is also the case for adolescents’ self-injury. Qualitative research found that the main self-injury motives of Korean adolescents were repeated conflicts with parents (family violence and parents’ dismissive attitude) and maladaptation to school life (academic pressure and conflicts with friends) [14]. In the past, academic stress was understood as a feeling high school students often experience while preparing for the college entrance exam, but even children at the senior years of elementary school suffer from learning issues as preschool education and prior learning become prevalent in Korea [15]. Many Korean parents’ interest and support or expectation and demands for children’s learning and school grades are immense. This may have a positive influence on children’s education, but at the same time may cause conflicts with children as well [12]. Adolescents spend most of their time at school with peers. During the middle and high school period, they develop a sense of identity while building friendships, thus having positive peer relationships is very important. However, those who have a difficulty in socializing with peers or experience bullying are more likely to attempt self-injury and suicide [16,17]. Since the poor socioeconomic environment is also highly likely to have adolescents exposed to negative incidents, the risk of self-injury according to family’s economic status needs to be evaluated. Little studies, however, reported the gender differences in the factors associated with NSSI in Korea.

Suicide is a significant problem for South Korean adolescents and is worse among females. Because the majority of adolescents’ lives exist either in school or at home, it is appropriate to evaluate NSSIs by sampling adolescents from across school environments. Therefore, this study was conducted to estimate the prevalence of NSSIs and to evaluate the related factors with NSSIs in multiple schools in a single city in Gyeonggi Province, near Seoul Metropolitan city, and analyses were stratified for gender.

## 2. Materials and Methods

### 2.1. Study Participants and Data Collection

This study’s subjects were recruited from all middle (12–14 years old) and high schools(15–17 years old) in a single city(A) in Gyeonggi Province, South Korea. Data regarding the number of classes and students at those schools were provided by the Office of Education in city A. In 2019, there were 20 middle and high schools in city A, with 505 classes in total (12 middle schools, 260 classes; eight high schools, 245 classes). A presentation for the survey was made for the school nurses, and each school’s participation was subsequently checked to increase the participation rate. One high school refused to participate in the investigation (Figure 2). 

The cluster-random sampling method was used to increase the efficiency. The investigation unit was a class; every school’s data were stratified based on the school year. A class was randomly selected each school year, and every student in the selected class was investigated. Notably, to consider the difference in class ratio between the middle and high schools, two classes were selected for high schools with more than ten classes. The final number of selected schools, classes, and students were as follows: 12 middle schools, 36 classes, and 1074 students, and seven high schools, 35 classes, and 873 students. In total, 19 schools, 71 classes, and 1947 students were selected as samples. All 71 classes participated in the investigation, and the final number of respondents was 1843, excluding the absentees on the day of the investigation (94.7% response rate) (Figure 1). 

The study period was 18 days from 2 July to 19 July, 2019: Each school chose the investigation date based on the academic calendar (final exams and vacations). The investigation was conducted using a self-administered online questionnaire. The students completed the questionnaire in the computer lab at 13 schools and used a tablet PC in the classroom at six schools. Few days before the investigation date, the code in the guideline sent to each school was entered into the computer before the investigation to protect each respondent’s personal information. Furthermore, the teachers supervised the students according to the guideline, ensuring they were in a position where the computer screen remained invisible. The time taken for completing the questionnaire was 20–30 min.

The survey in this study was conducted by a research agency Macromill Embrain through commissioning, with the approval of the Institutional Review Board at Hallym University (Approval No. HIRB-2019-022).

### 2.2. Measures 

The dependent variable in this study was the single question on NSSI; “Have you, in the recent past 12 months, had an episode of self-injury without any intention of committing suicide?” The participating students were asked to respond with yes or no. For the uniformity with the suicide-related questions (suicidal ideation, suicidal plans, and suicidal attempts), the period was restricted to the past 12 months. The NSSI questions in this study do not satisfy the DSM-5.

The main independent variables were the school year (Middle school Year 1–High school Year 3), self-rated school grades (High, Medium-high, Medium, Medium-low, Low), self-rated economic status (High, Medium-high, Medium, Medium-low, Low), suicidal ideation (No, Yes), suicidal plans (No, Yes), suicidal attempts (No, Yes), gaming disorder (No, Yes), and depressive feelings (No, Yes). Except for gaming disorder, all questions were based on the 2019 Korea Youth Risk Behavior Survey of the Korea Disease Control and Prevention Agency. The respective questions were as follows: For suicidal ideation, “Have you, in the past 12 months, had a day when you seriously considered committing suicide?”; For suicidal plans, “Have you, in the past 12 months, made a detailed plan to commit suicide?”; For suicidal attempts, “Have you, in the past 12 months, attempted to commit suicide?”; For depressive feelings, “Have you, in the past 12 months, ever felt so sad or hopeless that you could not continue daily activities for two whole weeks?”. The questions to measure gaming disorder were based on the World Health Organization’s (WHO) criteria; the loss of ability to control gaming, prioritizing gaming over other daily activities, and gaming in continuous excess despite the adverse effects. Any one of these symptoms defined the respondent as having a gaming disorder. For economic status, “How is your family’s economic status?” with responses of “High” “Medium-high” “Medium” “Medium-low” and “Low.” For school grades, “How was, in the past 12 months, your school grades” with responses of “High” “Medium-high” “Medium” “Medium-low” and “Low.” 

Current smoking was defined as “smoking even one cigarette (tobacco) in the recent 30 days.” Current drinking is defined as “drinking more than a glass of alcohol in the recent 30 days.” Physical activities are defined as “doing physical activities (that cause a rapid heartbeat or breathlessness) for more than 5 days, regardless of activity types for more than 60 minutes a day, in the recent 7 days.” The question to the bullying victims is “Have you felt, in the recent 30 days, distressed due to bullying or harassment by peers” with responses of “Yes” or “Seriously.”

### 2.3. Statistical Analysis

The gender distribution of study participants (Table 1) is presented as the percentage of students (%) through frequency analysis. Chi-square test of suicidal ideation, suicidal plans, and suicidal attempts according to NSSI was performed (Table 2). For the prevalence rate of NSSI for different gender characteristics, Chi-square test was performed (Figure 3). For the NSSI related factors, logistic regression analysis was performed. The effect of each variable as the primary predictor on the risk of NSSI was evaluated in the regression models, mutually adjusted for gender, school level, economic status, bullying victim, gaming disorder, and depressive symptom (Table 3). Stratified analysis by gender was performed with the same covariates (Table 4). For economic status, the two responses with a small number of respondents, “medium-low” and “low,” were merged. All analyses were carried out on gender stratification, for which the IBM SPSS Statistics ver. 25.0 (IBM, Armonk, NY, USA) was used at the significance level of 0.05.

## 3. Results

The number of the participating schools were 19, with 71 classes and 1843 students (1031 students at middle schools, 812 students at high schools; 968 male and 879 female students) (Figure 2). 

As shown in Table 1, the 12-month prevalence of NSSI was 8.8%; the rate was 2.85 times higher for female students (13.4%) than male students (4.7%). When applying the rates of NSSI by gender-age groups of middle and high school students to Korean teenagers aged between 12 and 17 (2,844,578 persons) in 2019, a total of 250,323 adolescents (male: 69,246 and female: 183,749) appeared to have tried self-injury for the last one year (data not shown). Those with suicidal ideation were 12.9% (male 8.0%, female 18.2%), suicidal plan 4.4% (male 2.9%, female 6.1%), and suicidal attempts 3.2% (male 1.1%, female 5.5%). Furthermore, depressive symptoms were higher for females than males (34.2% vs. 20.9%) while gaming disorder was higher for males than females (14.6% vs. 9.7%). The response “high” to economic status was higher for males(15.8%) than for females (9.3%). There were no gender differences in school level, school year, school grades, and bullying victims. 

The relationship between NSSI and suicidal ideation, suicidal plans, and suicidal attempts was summarized in Table 2. Among those experienced NSSI (*n* = 163), the percentages of students who had suicidal ideation, suicidal plans, and suicidal attempts in the recent 12 months were 54.6% (*n* = 89), 28.8% (*n* = 47), and 25.2% (*n* = 41), respectively. The percentage of suicidal ideation, suicidal plans, and suicidal attempts in the recent 12 months among female teenagers who experienced NSSI were 60.2%, 31.4%, and 30.5%, respectively, which is higher than male teenagers with 40.0%, 22.2%, and 11.1% (all of *p*-value < 0.001).

For both genders, the prevalence of NSSI was higher for middle school students compared to high school students(10.4% vs. 6.9%), as shown in Figure 3. The gender difference became apparent. Female students (middle school 15.8% and high school 10.2%) showed a significantly higher rate than male students (middle school 5.1% and high school 4.1%). For the school year, the highest rate was shown by Year 2 at middle school for female students (19.3%) and Year 3 at middle school and Year 1 at high school for male students (7.2%). 

The higher risk for NSSI was observed among female (multivariable-adjusted odds ratio [mOR] 2.73, 95% confidence interval [CI] 1.87–3.99), middle school (mOR 1.48, 95% CI 1.03–2.13), lower economic status (mOR 2.44, 95% CI 1.13–5.28), and those with gaming disorder (mOR 1.87, 95% CI 1.20–2.92) (Table 3). Gender-stratified analysis was shown in Table 4. For the school grade, mOR was 3.89 times (95% CI 1.35–11.26) higher for the group with the highest school records compared with the middle group for male students, while such pattern was not observed among female students (mOR 1.11, 95% CI 0.54–2.26). The risk for NSSI were higher among female students with lower economic status, compared to those with higher economic status (mOR 4.16, 95% CI 1.43–12.05), when economic status did not implicate higher risk among male students (mOR 1.43, 95% CI 0.42–4.84).

Gaming disorder appeared to be related with NSSI (mOR 1.87, 95% CI 1.20–2.92) with no material difference by gender (mOR 1.97, 95% CI 0.97–4.02 for male; mOR 1.92, 95% CI 1.07–3.41 for female). Those with depressive symptoms were stronly more likely to experience NSSI for both male (mOR 4.48, 95% CI 2.35–8.54) and female (mOR 6.11, 95% CI 3.90–9.57) students.

## 4. Discussion

The observed 12-month prevalence of NSSI was 8.8% and revealed strong differences according to gender (male: 4.7%; female: 13.4%) among Korean adolescents in this study. The summarized prevalence of NSSI was reported to be 19.5% from a recent meta-analysis [8] and 15.0%-29.2% from studies for students in the U.S. and other countries [18,19,20,21,22]. 

In comparison with other Korean studies, the study results were similar to 7.9% and 6.4% of the 3-month prevalence of NSSI for first-year middle school students and first-year high school students, respectively, as studied by the Ministry of Education in 2018 [6] and 14.3% of the lifetime prevalence of NSSI of female middle school students in Busan [23]. The study results were lower than 13.8% in the 12-month prevalence of NSSI for middle and high school students in three Jeollanamdo cities/regions [24], 13.8% of the 12-month prevalence of NSSI for male students in five specialized high schools in Incheon [25] and 22.8% for middle and high school students in the Youth Dream Up Center (Delinquency Prevention Center) operated by Seoul, Gyeonggi-do and the Ministry of Justice [26]. The Ministry of Education study is significant as the first report on the prevalence of adolescent self-injury. Nonetheless, as it focused on self-injury experience within three months, it is differentiated from other studies that generally focused on lifetime or 12-month prevalence of NSSI. Moreover, the study conducted on male students in a vocational high school in Incheon city in the same period as this study (July 2019) showed a 3.4 times higher 12-month prevalence of NSSI (13.8%) than that for male high school students in this study (4.1%) and a 2.2 times higher prevalence than that for Year 1 high school students in the study by the Ministry of Education (6.4%). The high prevalence was attributed to the high proportion of students economically vulnerable as the target school was a vocational high school [25]. The higher NSSI prevalence in this study from Ahn, Y.S et al. [26] can be explained by the fact that it included students from a specialized school called the Youth Dream Up Center.

The higher prevalence of NSSI among female students observed in this study is consistent with previous studies that showed that the prevalence of NSSI in female adolescents was 1.8 to 2.4 times higher than that of males [18,19,20].

Bresin et al. conducted a meta-analysis and reported that the prevalence of NSSI was 1.5 times higher for females than males (95% CI 1.35–1.65) [27,28]. Valencia-Agudo et al. analyzed 39 articles of prospective studies on adolescents in regional communities [29]. They showed that gender (female) was the most potent predictor of NSSI among the sociodemographic factors. Female adolescents are more vulnerable to negative emotions (depression, stress, anger, etc.), which are the most well-known risk factors of self-injury [9,30]. The subjects in this study had depressive symptoms 34.2% in females and 20.9% in males, while the awareness of stress was higher in females (48.5%) than males (32.0%). In South Korea, female adolescents make more suicidal behaviors. Among the subjects of this study, female students reported to have over two times more suicidal ideation, suicidal plans, and suicidal attempts. There is also a report that the number of female adolescents moved to the emergency room due to self-injury and suicidal attempts is higher than male adolescent [31]. Notably, 30.5% of female students who reported NSSI in this study had attempted suicide during the same period. This is three times higher than of male students (11.1%). It demonstrates that there is a closer relationship between NSSI and suicidal behaviors in females than males, despite no intention to commit suicide. It also represents that the rapid increase of suicide rates among Korean female adolescentsteenage girls may be attributable to the high prevalence of NSSI in female adolescents.

Middle school students were 1.68 times higher risk for NSSI than high school students, while such school level difference was not observed for male students (Table 4), indicating earlier start of NSSI experience among teen girls. Self-injury has been reported to start in childhood and early adolescence (12–14 years) and peak in mid-adolescence (15–16 years) with a subsequent fall in late adolescence (~18 years) [10,23,25,32], sometimes continuing into early adulthood [32]. A previous study in South Korea reported that self-injury first occurs at the age of 12.4 years for females [23], which is 1–2 years earlier than 13.9 years for males [25], which is consistent with our findings. The age at which the highest prevalence of self-injury was observed among adolescents in this study was also earlier for female students (Year 2 middle school; 14 years) than males (Year 3 middle school - Year 1 high school; 15–16 years). In sum, for male students, a gradual, normal distribution starting at middle school and ending at high school was found. For female students, the distribution started earlier with a subsequent steep increase, which continued into early adulthood.

In male students, highest school grades were the risk factor of NSSI. Previous studies reported that, while low school grades and academic stress influence self-injury (suicide), the influence is not direct but requires a medium of negative emotions (depression, anxiety, helplessness, anger, etc.) [30,33,34]. Further, it is not the school grades themselves but the excess burden of studying and academic stress is accountable for higher rate of self-injury (suicide) [30]. Jun and Choi [15] investigated the impact of school grades on the sense of happiness parents and children feel and found that children’s improved school grades increased a sense of happiness for both children and parents, but high-performing students were less happy. Based on these reports, within the educational system in South Korea with a focus on university admission, students with high school grades may be under more significant academic pressure than students with low school grades. Moreover, their reputation as an ideal student and their friends and family’s high expectations drove these students to choose self-injury to relieve the academic stress secretly. In the Avon Longitudinal Study of Parents and Children (ALSPAC), a prospective cohort study conducted in the U.K. based on population, a higher correlation between high IQ and NSSI risk was found for both males than females in 4810 adolescents aged 16–17 years [35]. Although a direct comparison based on school grades and IQ is challenging, one may hypothesize that male students with high school grades would be more sensitive to academic stress than female students. This study did not find whether there was a gender difference in academic pressure among high-performing students; otherwise there is a gender difference in the way of relieving stress from academic pressure. Further studies should be conducted with a well-structured design to prove such a hypothesis.

Economic status was a related factor of NSSI in female students only, with a lower economic status indicating a higher risk of NSSI. This is a result that supports the preceding studies [25,36] that NSSI is higher among adolescents from socioeconomically vulnerable families. They did not directly deal with the relationship between poverty and NISS, but Kim E.I and Oh K.J [37] provide evidence that proves the relationship between female adolescents from poor families and high NSSI. Kim and Oh [37] investigated the gender difference for middle school students in poverty and sensitivity to stress. A one-year follow-up found that female adolescents from poor families are more sensitive to stressful life events than male adolescents and that female students become more vulnerable to a sense of depression and problematic behaviors than male students when stress increases. Park et al. conducted a study on the correlation between suicide ideation in adolescents and their parent’s socioeconomic status [38]. Female students were more sensitive to the parent’s socioeconomic status than male students [38]. Moreover, female adolescents from poor families are more likely to join the labor market in their early ages and to give up college admission than male adolescents in Korea according to the report of National Youth Policy [39]. A more careful gender-sensitive approach is needed for effective interventions for the self-injury of adolescents from poor families.

The limitations of this study are as follows. First, this study’s results cannot be generalized to all middle and high school students in South Korea, as the subjects were middle and high school students in a single city. However, considering that the results were similar to the results of the study conducted by the Ministry of Education on all middle and high schools in South Korea, the prevalence of and related factors with NSSI found in this study may be taken to represent those of the middle and high school students. Additionally, random sampling was performed on all middle and high schools in the study area, while the selected subjects’ response rate was 94.7%, indicating a high internal validity. Second, the advantage of schools is easy contact with study subjects. However, because the survey was conducted in a school setting, the participants may have found it difficult to provide honest responses. The participants were guaranteed voluntary participation and anonymity to minimize such information bias. With a guideline, the teachers were instructed to supervise the students, not to see the students’ responses. Third, this study used a single question on self-injury, which does not satisfy the DSM-5 [7]. Consequently, the information related to the number of self-injury episodes in the past year, the method of self-injury, or the intensity of self-injury could not be obtained. Also, responders may have answered based on the latest self-injury episodes of self-injury because the question was about the self-injury attempted for the past 12 months. If so, the rate of self-injury in this study might have been underestimated. However, by clearly stating that the self-injury in concern is the one without the intention to commit suicide, the participants were informed of the differentiation from suicidal attempts.

Furthermore, the questionnaire was composed to allow the main independent variables (school grades, economic status, etc.) to be subjectively rated by the respondent. It is thus possible that actual circumstances had not been reflected. As school grades or economic status are sensitive topics, a high response rate cannot be expected. Therefore, a large-scale study for trend analysis should be conducted, for which the self-rated questions are predicted to be suitable. Finally, we did not ask any questions on sexual preference in this survey. We may not identify the LGBTQIA subgroup with higher risk for NSSI than binary gender [40].

## 5. Conclusions

The gender-specific trend based on the distribution of prevalence across male and female students was such that, for male students, a gradual, normal distribution starting at middle school and ending at high school was found. For female students, the distribution started earlier with a subsequent steep increase, which continued into early adulthood. The related factors of NSSI varied according to gender when multivariate logistic regression analysis was performed. For male students, the risk of self-injury was high in the group with high school grades, but for female students, the risk increased in middle school students and the group with low economic status.

This study suggests that the risk for NSSI among adolescent student may be different according to gender and comprehensive interventions from schools against NSSI may need a gender-sensitive approach in South Korea. Based on this study’s findings, further studies should be conducted using a well-structured, nationwide design to determine more accurate representation and risk factors regarding self-injury among South Korean adolescents.

## Figures and Tables

**Figure 1 ijerph-18-05965-f001:**
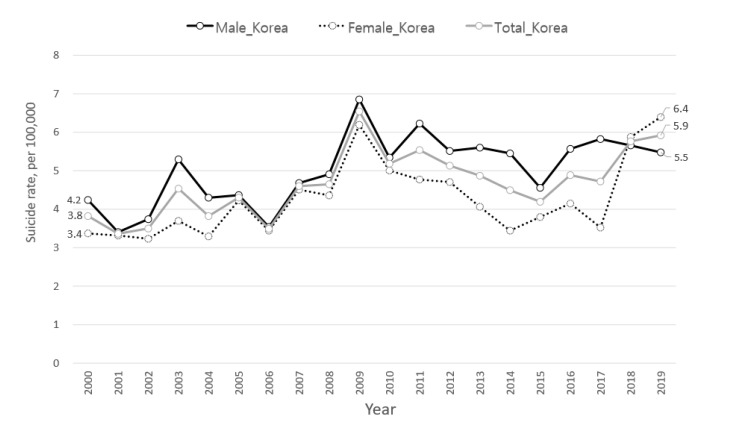
Trend of suicide rates among 10–19 years old in South Korea.

**Figure 2 ijerph-18-05965-f002:**
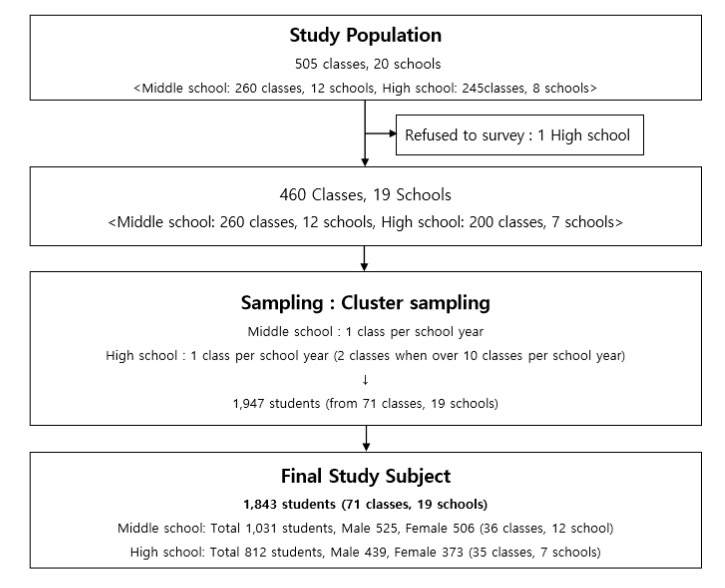
Prevalence survey for non-suicidal self-injury in South Korea, 2019.

**Figure 3 ijerph-18-05965-f003:**
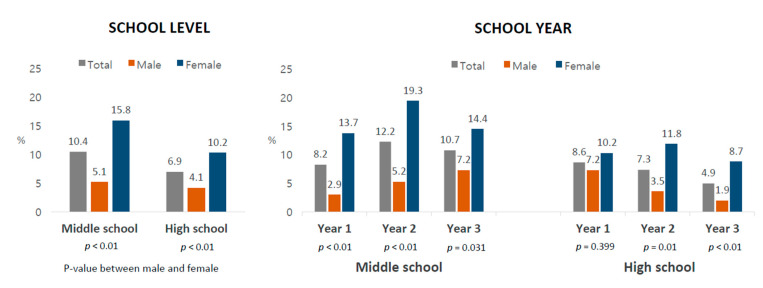
12-month prevalence of non-suicidal self-injury by gender among adolescent students in South Korea, 2019.

**Table 1 ijerph-18-05965-t001:** General Characteristics of study subjects of prevalence survey for non-suicidal self-injury among adolescents in South Korea, 2019.

Variables /Classification	Total	Male	Female
	*n* = 1843 (%)	*n* = 964 (%)	*n* = 879 (%)
School level			
Middle school (12–14 years old)	1031 (55.9)	525 (54.5)	506 (57.6)
High school (15–17 years old)	812 (44.1)	439 (45.5)	373 (42.4)
School year			
Middle school Year1 (12 years old)	340 (18.4)	172 (17.8)	168 (19.1)
Year2 (13 years old)	344 (18.7)	173 (17.9)	171 (19.5)
Year3 (14 years old)	347 (18.8)	180 (17.8)	167 (19.0)
High school Year1 (15 years old)	266 (14.4)	138 (14.3)	128 (14.6)
Year2 (16 years old)	262 (14.2)	143(14.8)	119 (13.5)
Year3 (17 years old)	284 (15.4)	158 (16.4)	126 (14.3)
School grades			
High	249 (13.5)	149 (15.5)	100 (11.4)
Medium-high	486 (26.4)	241 (25.0)	245 (27.9)
Medium	567 (30.8)	284 (29.5)	283 (32.2)
Medium-low	382 (20.7)	201 (20.9)	181 (20.6)
Low	159 (8.6)	89 (9.2)	70 (8.0)
Economic status			
High	234 (12.7)	152 (15.8)	82 (9.3)
Medium-high	565 (30.7)	292 (30.3)	273 (31.1)
Medium	884 (48.0)	432 (44.8)	452 (51.4)
Medium-low/Low	160 (8.7)	88 (9.1)	72 (8.2)
Current smoking. Yes (%)	97 (5.3)	75 (7.8)	22 (2.5)
Current drinking, yes (%)	188 (10.2)	120 (12.4)	68 (7.7)
Physical activity for 60min a day and 5 days or over a week, yes (%)	284 (15.4)	223 (23.1)	61 (6.9)
Non-suicide self-injury, yes (%)	163 (8.8)	45 (4.7)	118 (13.4)
Suicidal ideation, yes (%)	237 (12.9)	77 (8.0)	160 (18.2)
Suicidal plan, yes (%)	82 (4.4)	28 (2.9)	54 (6.1)
Suicidal attempts, yes (%)	59 (3.2)	11 (1.1)	48 (5.5)
Depressive symptom, yes (%)	502 (27.2)	201 (20.9)	301 (34.2)
Bullying victim, yes (%)	43 (2.3)	24 (2.5)	19 (2.2)
Gaming disorder, yes (%)	226 (12.3)	141 (14.6)	85 (9.7)

**Table 2 ijerph-18-05965-t002:** Suicidal ideation, suicidal plan, and suicidal attempts rates according to non-suicidal self-injury by gender.

Non-Suicidal Self-Injury	Total			Male			Female		
	No (*n* = 1680)	Yes (*n* = 163)	*p*-Value	No (*n* = 919)	Yes (*n* = 45)	*p*-Value	No (*n* = 761)	Yes (*n* = 118)	*p*-Value
Suicidal ideation No Yes	1532 (91.2) 148 (8.8)	74 (45.4) 89 (54.6)	<0.010	860 (93.6) 59 (6.4)	27 (60.0) 18 (40.0)	<0.01	672 (88.3) 89 (11.7)	47 (39.8) 71 (60.2)	<0.01
Suicidal plan No Yes	1645 (97.9) 35 (2.1)	116 (71.2) 47 (28.8)	<0.01	901 (98.0) 18 (2.0)	35 (77.8) 10 (22.2)	<0.01	744 (97.8) 17 (2.2)	81 (68.6) 37 (31.4)	<0.01
Suicidal attempts No Yes	1662 (98.9) 18 (1.1)	122 (74.8) 41 (25.2)	<0.01	913 (99.3) 6 (0.7)	40 (88.9) 5 (11.1)	<0.01	749 (98.4) 12 (1.6)	82 (69.5) 36 (30.5)	<0.01

**Table 3 ijerph-18-05965-t003:** Related factors with non-suicidal self-injury among adolescents in South Korea.

Variables/Category	NSSI/No NSSI	Model 1	Model 2	Model 3
		OR (95% CI)	OR (95% CI)	OR (95% CI)
Gender				
Male	45/919	1.0	1.0	1.0
Female	118/761	3.14 (2.20–4.49)	3.49 (2.41–5.03)	2.73 (1.87–3.99)
School level				
High school	56/756	1.0	1.0	1.0
Middle school	107/924	1.53 (1.09–2.15)	1.47 (1.03–2.08)	1.48 (1.03–2.13)
School grades				
High	27/222	1.56 (0.93–2.62)	1.74 (1.02–2.98)	1.67 (0.95–2.92)
Medium-High	38/448	1.00 (0.63–1.58)	1.04 (0.65–1.66)	1.08 (0.67–1.76)
Medium	43/524	1.0	1.0	1.0
Medium-Low	37/345	1.37 (0.86–2.19)	1.31 (0.81–2.10)	1.16 (0.71–1.90)
Low	18/141	1.77 (0.98–3.20)	1.48 (0.80–2.75)	1.14 (0.60–2.16)
Economic status				
High	14/220	1.0	1.0	1.0
Medium-High	51/514	1.37 (0.74–2.55)	1.63 (0.86–3.09)	1.80 (0.93–3.49)
Medium	74/810	1.26 (0.69–2.29)	1.45 (0.77–2.72)	1.50 (0.79–2.86)
Medium-Low/Low	24/136	2.71 (1.34–5.50)	2.78 (1.33–5.85)	2.44 (1.13–5.28)
Bullying victim				
No	153/1347	1.0	1.0	1.0
Yes	10/33	3.39 (1.60–7.21)	3.34 (1.55–7.22)	1.79 (0.81–3.93)
Gaming disorder				
No	127/1490	1.0	1.0	1.0
Yes	36/190	2.58 (1.71–3.91)	2.14 (1.60–3.71)	1.87 (1.20–2.92)
Depressive symptom				
No	53/1288	1.0		1.0
Yes	110/392	6.14 (4.32–8.73)		5.48 (3.81–7.89)

Model 1: Adjusted for gender, school level; Model 2: Adjusted for Model 1 + school grades, economic status, bullying victim and gaming disorder; Model 3: Adjusted for Model 2 + depressive symptom.

**Table 4 ijerph-18-05965-t004:** Gender differences in related factors with non-suicidal self-injury among adolescents in South Korea.

Variables/Category	Male				Female			
	NSSI/No NSSI	Model 1 OR(95% CI)	Model 2 OR(95% CI)	Model 3 OR(95% CI)	NSSI/NO NSSI	Model 1 OR(95% CI)	Model 2 OR(95% CI)	Model 3 OR(95% CI)
School level								
High school	18/421	1.0	1.0	1.0	38/335	1.0	1.0	1.0
Middle school	27/498	1.27(0.69–2.34)	1.18(0.62–2.26)	1.19(0.62–2.30)	80/426	1.66 (1.10–2.50)	1.61 (1.05–2.46)	1.68 (1.08–2.61)
School grades								
High	12/137	3.89 (1.42–10.66)	4.00 (1.41–11.41)	3.89 (1.35–11.26)	15/85	1.07 (0.56–2.06)	1.22 (0.62–2.40)	1.11 (0.54–2.26)
Medium-High	9/232	1.75 (0.61–5.00)	1.71 (0.59–4.95)	1.62 (0.56–4.74)	29/216	0.97 (0.52–1.46)	0.93 (0.55–1.58)	1.01 (0.58–1.78)
Medium	6/278	1.0	1.0	1.0	37/246	1.0	1.0	1.0
Medium-Low	13/188	3.26 (1.22–8.74)	3.13 (1.14–8.57)	2.52 (0.91–6.96)	24/157	1.03 (0.59–1.78)	0.95 (0.54–1.67)	0.86 (0.47–1.55)
Low	5/84	2.89 (0.85–9.76)	2.89 (0.83–10.11)	2.43 (0.68–8.65)	13/57	1.57 (0.78–3.16)	1.23 (0.59–2.56)	0.88 (0.40–1.90)
Economic status								
High	8/144	1.0	1.0	1.0	6/76	1.0	1.0	1.0
Medium-High	17/275	1.12 (0.47–2.65)	1.44 (0.58–3.60)	1.50 (0.59–3.78)	34/239	1.85 (0.75–4.59)	2.16 (0.85–5.46)	2.51 (0.96–6.54)
Medium	14/418	0.62 (0.25–1.51)	0.82 (0.32–2.12)	0.81 (0.31–2.14)	60/392	2.04 (0.85–4.91)	2.29 (0.93–5.64)	2.51 (0.99–6.36)
Medium-Low/Low	6/82	1.40 (0.46–4.22)	1.55 (0.47–5.08)	1.43 (0.42–4.84)	18/54	4.44 (1.65–11.96)	4.47 (1.61–12.43)	4.16 (1.43–12.05)
Bullying victim								
No	41/899	1.0	1.0	1.0	6/13	1.0	1.0	1.0
Yes	4/20	4.35 (1.42–13.33)	3.86 (1.21–12.37)	2.07 (0.61–6.97)	112/748	2.81 (1.04–7.60)	2.93 (1.04–8.30)	1.61 (0.56–4.61)
Gaming disorder								
No	32/791	1.0	1.0	1.0	95/699	1.0	1.0	1.0
Yes	13/128	2.47 (1.26–4.85)	2.39 (1.20–4.78)	1.97 (0.97–4.02)	23/62	2.67 (1.58–4.52)	2.54 (1.48–4.36)	1.92 (1.07–3.41)
Depressive symptom								
No	20/743	1.0		1.0	33/545	1.0		1.0
Yes	25/176	5.319 (2.87–9.90)		4.48 (2.35–8.54)	85/216	6.59 (4.27–10.17)		6.11 (3.90–9.57)

Model 1: Adjusted for school level; Model 2: Adjusted for Model 1 + school grades, economic status, bullying victim and gaming disorder; Model 3: Adjusted for Model 2 + depressive symptom.

## Data Availability

The data presented in this study are available on request from the corresponding author.

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
