# Peer review of "Gender Differences in the Prevalence of and Factors Related to Non-Suicidal Self-Injury among Middle and High School Students in South Korea"

_ijerph, 2021, doi:10.3390/ijerph18115965_

Round 1

Reviewer 1 Report

Dear Author(s),

It is very intresting text, congratulation. I admire  your survay. In Poland - I predicted - there are a lot of problems doing survay like that, because of  school reluctance.

My questions:

  1. The numer of suicide among adolescents in 2021- but what about long term perspective? Are there any data about earlier tendency in Korea? For instance 2010-2019.
  2. NSSI among adolesctents (19,5%) but have You got any quantitative data- how many people are there in Korea? 100 thousend, 200, because that percpestive sticks with country
  3. The main causes of NSSI there are in text, but are any prospect to describe that problem NSSI and your results accordance any theories?
  4. Figure 1- could You explain motivation of refusal participation in survay? The point is- did the school argument that decision in ethical aspects (any ethical doubts?)?
  5. Tabele 1- I don’t understand the different between "suicidal ideation" and "suicidal plan"? Ideation (fantasy about self-harm) and the plan. The point is- are You sure, that participants understood the differences in test?
  6. The situation of students in educational system- As educationalist, I regret, that problem was described only in one view- only presure.
  7. The future hypothesis- male students in high school are more sensitive to academic stress- but what does it mean? Are there any evidences- it could be true? May there is something in Korea (micro or macro socio), and it makes an impact on male students before adults? It is very interesting byt I don’t understand the basic of hypothesis, sorry.
  8. The limitations of survay are very clear.

Reviewer 2 Report

Review for manuscript entitled ‘Gender differences in the Prevalence of and Factors Related to non-Suicidal Self-Injury among Middle or High School Students in South Korea’

Using cluster-random sampling, the authors estimated the prevalence of NSSI and explored gender differences in middle/high school students in South Korea. The data is valuable and the sample size was quite big N=1843. As South Korea has been one of the high suicide rates countries in the world, data on NSSI are very much needed.

However, the study is not hypotheses driven; the organization of the manuscript is too diffuse, not very focused. The authors have to address hypotheses in the introduction section (e.g. what do you think school grade/year would affect NSSI? What is the hypothesized impact of school grade/year by gender? Why you assumed there were gender differences? The discussion section would be more focused on the hypotheses you proposed. It is probably more reasonable to do prevalence of NSSI, and to explore factors related to NSSI and then test whether these factors interacted with gender. If the interaction effect was significant, then you go on to do gender stratified analysis.

Table 1: in addition to NSSI, you listed suicide attempt, and suicide plan do you think these variables would co-vary with NSSI? No explanations about why you listed these variables in introduction. You might need to let the readers know how many individuals with NSSI had history of suicide ideation/plan/attempt. NSSI, individuals with suicidal ideation/plan/attempt may be overlapping populations. Probably it is a good idea to list distributions of study variables according to outcome of interest. In other words, the table columns would contain NSSI/no-suicidality/suicidality (suicide idea/attempt/plan).

Figure 2: why gender differences in year 1 of high school was so small? Please explain.

Figure 3. The differences in SES by gender, what is the explanation? Did you expect to see this? (i.e. you have to advance your hypothesis in introduction and explain your findings).

Please provide a table for your multivariate analyses.

Discussion: please explain the meaning of your findings. The current version was just comparing the numbers with previous research. Please provide ‘meaning’ and provide readers take home messages of this article.

Reviewer 3 Report

Summary

This manuscript presents the results of a survey study of South Korean adolescents on NSSI, suicidal thoughts and behaviors, and other clinical and sociodemographic data. While the prevalence estimates of NSSI are useful, they are not novel when compared to what is already known. More complicated statistical analyses that could present novel findings (e.g., relationships with grades, within school levels) are hampered by unclear reporting of the methodology, significant limitations of single-item self-report 12-month recall questions in adolescents, and a poorly focused discussion that vastly exceeds the study data. Consequently, the study does not offer novel information in a clear or concise way.

Major comments

  • While I think this study is useful and purposeful, this purpose is not adequately established in the introduction. The authors state that “the accurate, current status of adolescent self-injury has not been investigated”, but they cited a worldwide meta-analysis of NSSI (reference 7) which is as recent 2018. One way to address this is to narrow the focus to just South Korean adolescents and the gender differences as they relate to grades (which was a major finding in the study).
  • On a related note, there are no hypotheses stated in the introduction. What did they authors want to investigate and what did they expect to find?
  • As far as I can tell, there is no results section in this paper. The numbering goes from Section 2 to Section 4. I think the results are presented on page 6/13, but it’s not immediately clear.
  • The statistical analysis is unclear: logistic regressions were used, but was there only 1 model that included all of the independent variables specified? Or were there multiple models with only a single IV in each model? Incidentally, the supplementary table should be in the main manuscript and Figure 3 should be removed since reporting odds ratios in tables with 95% confidence intervals is much clearer than bar graphs.
  • The discussion is really unfocused and makes it hard to determine what the significant findings of the study are. A lot of the articles reviewed in the discussion should have been introduced in the introduction to help set up what the current knowledge about NSSI is in South Korea and what the knowledge gaps are. As it is, these references showing up in the discussion undermines the author’s contention that little is known about adolescent NSSI.
  • The discussion also involves a very long and very speculative discussion about all the reasons why their findings are they way they are. For example, the heightened prevalence of NSSI in females is attributed to South Korean patriarchal culture. These explanations are inappropriate because they ignore more obvious explanations (e.g., females all over the world are more likely to have NSSI than males, so South Korean culture seems irrelevant), and because they cannot be justified in light of the significant limitations of the paper, namely the single-item measures, the self-report data, the potential for recall bias, and the potential for underreporting, which are all exacerbated in an adolescent population. The discussion should be edited/revised to narrowly focus on the results at hand with limited speculation about potential explanations.
  • It seems the first discussion point is about how the prevalence of NSSI in this study was less than in other studies, and actually is similar to a 3-month prevalence estimate of NSSI. To me, this basically reads that students, when asked about their NSSI in the last year, just remembered what happened in the last 3 months and reported that. That’s a major limitation of single-item retrospective self-report questions, especially in minors who are unlikely to recall their experiences accurately.

Minor comments

  • There are significant English language grammar issues that should be corrected, including a rather gratuitous use of semicolons.
  • There are a lot of unsubstantiated claims in this paper (e.g., suicide is the most severe mental health problem in South Korea, or NSSI is viewed as mere delinquency). These opinions should either be substantiated with citations or removed/replaced with opinion neutral statements.
  • It would be useful to have the actual suicide rate for adolescents 10-19 in the paper, not just the percent of mortality attributable to suicide.
  • There is a statement in the introduction, paragraph 4, about how NSSI has an “addictive nature”. This comment is not elaborated on or substantiated, so I’m not sure what this means.
  • Intro Paragraph 5 uses an acronym (“SNS”) which I am unfamiliar with. I suspect it is a common social media system in South Korea, but it should be defined for the reader.
  • Is the name of the city literally “A”? Otherwise, I would just say the data were collected in Gyeonggi province South Korea.
  • Why was “gaming disorder” investigated? Was there an a priori hypothesis about how gaming disorder would affect NSSI?
  • The prevalence of depression symptoms was 27.2% in the sample; is this characteristic of South Korean adolescents? Or is this really high? This should be addressed.
  • I’m not sure Figure 2 adds any information that isn’t effectively conveyed in Table 1.
  • Prevalence estimates are reported on page 6 with statements that they differ by gender, but there are no statistical comparisons between genders.
  • I am unclear why some results are highlighted blue and underlined in the supplementary table. It doesn’t appear to be related to significance, so please clarify.
  • The authors claim that there is a U-shaped relationship between self-harm and grades in male students, but the associated with low grades is non-significant. Consequently, there is not a U-shaped relationship, just an increased prevalence for students with high grades.

Round 2

Reviewer 2 Report

Review for IJEPH manuscript entitled “Gender differences in the Prevalence of and Factors Related to Non-Suicidal Self-Injury among Middle and High School Students in South Korea”

Compared to the previous version, the current manuscript improved a lot. Specific comments are listed below—

  1. 2nd line of introduction, “it had peaked at 6.5 per 100000 in 2009….”, please indicate the age group for the statement. 6th line of introduction – “teenage girls was 6.4 per 100000, 1.9 times higher than the OECD average…”, what about boys? Please provide a comprehensive gender-specific picture.
  2. The structure of the introduction can be more focused – first paragraph; epidemiology of overall suicide trends/rates, 2nd paragraph: epidemiology of suicidal behaviours, 3rd paragraph: epidemiology of NSSI, 4th paragraph: factors associated with NSSI or suicidality, 5th paragraph what is lacking in previous research and the aim of your current research.

Your current version contains 9 paragraphs of introduction, too fragmented. Please condense a bit.

  1. Discussion – again, too long and too fragmented. Currently you have 11 paragraphs for discussion (not including conclusion). Please re-structure and make your discussion more focused (just focused on your key findings). The first paragraph of discussion should describe your main findings, and the rest of your discussion should then explain the meaning of your main findings. For example, the main findings of your discussion can present 1.) The observed prevalence of NSSI and briefly described whether the level of prevalence was expected, 2) Summarize correlates of NSSI such as the observed correlation with school level, school grades and gender differences (e.g. SES).
  2. SES and female sensitivity: you should contextualize your observation in South Korea. For example, in East Asian Confucianism culture, females are usually discriminated. In lower SES family, when family resources are limited, would it be possible that girls would be less likely to get family resources compared to boys? Or girls would be asked to share more family responsibilities, compared to boys (e.g. household chores). Their disadvantaged position might be related to their risk of suicide. Contextualize your observation is very important, you stated that girls are more sensitive to low SES, but it might be related to socio-cultural context that put girls (particularly girls in the low SES) at risk. Alternatively, it is possible that low SES boys express their stress in different ways, such as substance use, seeking for approval from peers (e.g. become gang members). Gender differences might be related to differences in ways of expressing stress.   

Author Response

REVIEWER 2

We greatly appreciate for your detailed and helpful comments. We revised our manuscripts as much as possible based on your suggestions.

Comments and Suggestions for Authors

Review for IJEPH manuscript entitled “Gender differences in the Prevalence of and Factors Related to Non-Suicidal Self-Injury among Middle and High School Students in South Korea”

Compared to the previous version, the current manuscript improved a lot. Specific comments are listed below—

1.  2nd line of introduction, “it had peaked at 6.5 per 100000 in 2009….”, please indicate the age group for the statement. 6th line of introduction – “teenage girls was 6.4 per 100000, 1.9 times higher than the OECD average…”, what about boys? Please provide a comprehensive gender-specific picture.

--> added as suggested

2. The structure of the introduction can be more focused – first paragraph; epidemiology of overall suicide trends/rates, 2nd paragraph: epidemiology of suicidal behaviours, 3rd paragraph: epidemiology of NSSI, 4th paragraph: factors associated with NSSI or suicidality, 5th paragraph what is lacking in previous research and the aim of your current research.

Your current version contains 9 paragraphs of introduction, too fragmented. Please condense a bit.

 --> revised as suggested, condensing 9 paragraphs to 5; 1. epidemiology of suicide in Korea, 2. epidemiology of suicidal behaviours, 3. epidemiology of NSSI,  4. related factors with NSSI,  and 5. study  objectives

3.            Discussion – again, too long and too fragmented. Currently you have 11 paragraphs for discussion (not including conclusion). Please re-structure and make your discussion more focused (just focused on your key findings). The first paragraph of discussion should describe your main findings, and the rest of your discussion should then explain the meaning of your main findings. For example, the main findings of your discussion can present 1.) The observed prevalence of NSSI and briefly described whether the level of prevalence was expected, 2) Summarize correlates of NSSI such as the observed correlation with school level, school grades and gender differences (e.g. SES).

 --> We deleted some paragraphs, including meta-analysis results in this discussion part, and tried to restructure and focus key findings. 

4. SES and female sensitivity: you should contextualize your observation in South Korea. For example, in East Asian Confucianism culture, females are usually discriminated. In lower SES family, when family resources are limited, would it be possible that girls would be less likely to get family resources compared to boys? Or girls would be asked to share more family responsibilities, compared to boys (e.g. household chores). Their disadvantaged position might be related to their risk of suicide. Contextualize your observation is very important, you stated that girls are more sensitive to low SES, but it might be related to socio-cultural context that put girls (particularly girls in the low SES) at risk. Alternatively, it is possible that low SES boys express their stress in different ways, such as substance use, seeking for approval from peers (e.g. become gang members). Gender differences might be related to differences in ways of expressing stress. 

--> added and revised as suggested

Moreover, female adolescents from poor family are more likely to join the labor market in their early ages [38]  and to give up college admission [39] than male adolescents in Korea according to the report of National Youth Policy. Adolescents from poor families are more exposed to factors affecting parenting such as parents’ depressive feelings and problematic drinking habits as well as economic problems than adolescents from non-poor families. Since the factors may intensify the conflicts between parents and children regardless of poverty [40], A more careful approach is needed to interventions for the self-injury of adolescents from poor families. 

Reviewer 3 Report

Summary

This revised manuscript presents the results of a cross-sectional analysis of sociodemographic and clinical predictors of non-suicidal self-injury among South Korean middle and high school students. The authors were clearly responsive to prior comments on their manuscript, and while some issues remain around communicating the study methodology, the manuscript is significantly stronger than its original submission.

Major Comments

  • I appreciate the authors’ willingness to extensively revise the introduction. As I understand it, the flow of the introduction is as follows (paraphrased): “Suicide is a significant problem for South Korean adolescents and is worse among females. Because the majority of adolescents’ lives exist either in school or at home, it is appropriate to evaluate NSSIs by sampling adolescents from across school environments. Therefore, NSSIs were evaluated in multiple schools and analyses were stratified for gender.” I think these changes improve the clarity and focus of the paper and prepare the reader for the results and discussion.
  • Because Tables 1-4 report findings from the sample, they should be discussed as part of the results section, not in the methods.
  • The logistic regression analyses are still not clearly explained in section 2.3 (statistical analysis), although Table 3 does help illuminate the approach being used. In my understanding, each variable in the Variables/Category column was the primary predictor in the logistic regression models, and then the models were adjusted for the additional variables mentioned in the table footnote (Model 1: gender + school level, Model 2: gender + school level + grades + economic status + bullying + gaming disorder, etc). However, this doesn’t quite make sense to me, since you can’t adjust for gender in a model where gender is the primary predictor. In explaining the statistical analysis, it should be clear what the primary predictor was (gender, school level, etc), and what, if any, additional covariates were included in that particular model.
  • That said, the data provided in Tables 3 and 4 are very helpful for understanding what is being analyzed and what the findings are.
  • English grammar continues to be a problem; while the text is not unreadable, spelling errors and grammatical mistakes detract from the clarity of the data and ideas presented.

Minor Comments

  • The authors cite something called the OECD, but this is not defined. All acronyms should be fully spelled out with the first use and then abbreviated thereafter.
  • There still aren’t any specific hypotheses about what the authors thought they would find in the introduction. I realize this seems pedantic since the text mentions stratifying analyses by gender, but stratifying by gender only makes sense if you expect a gender difference (which you do, given the relevant suicide data), and so your specific hypothesis would be that you expected females to have higher rates of NSSI than males. The point in mentioning this, by the way, is to disclose to the reader your expectation bias in conducting the study – you fully expected that females would have higher rates of NSSI and now the reader can assess if your data really supports that expectation.
  • I actually think it is appropriate to find that NSSI rates decrease significantly as students get older (i.e., from middle school to high school), as this follows observed data from around the world. NSSI peaks in adolescents and tapers off into adult hood except in seriously mentally ill populations, where self-harm becomes a staple of emotion regulation.

Reviewed by: Andrew S. Tubbs, MD/PhD Candidate, University of Arizona College of Medicine – Tucson.

Author Response

Reviewer 3

We greatly appreciate for your detailed and helpful comments. We revised our manuscripts as much as possible based on your suggestions.

Comments and Suggestions for Authors

Summary

This revised manuscript presents the results of a cross-sectional analysis of sociodemographic and clinical predictors of non-suicidal self-injury among South Korean middle and high school students. The authors were clearly responsive to prior comments on their manuscript, and while some issues remain around communicating the study methodology, the manuscript is significantly stronger than its original submission.

Major Comments

I appreciate the authors’ willingness to extensively revise the introduction. As I understand it, the flow of the introduction is as follows (paraphrased): “Suicide is a significant problem for South Korean adolescents and is worse among females. Because the majority of adolescents’ lives exist either in school or at home, it is appropriate to evaluate NSSIs by sampling adolescents from across school environments. Therefore, NSSIs were evaluated in multiple schools and analyses were stratified for gender.” I think these changes improve the clarity and focus of the paper and prepare the reader for the results and discussion.

Because Tables 1-4 report findings from the sample, they should be discussed as part of the results section, not in the methods.

  • Revised as suggested. Findings reported in the table 1-4 were moved to the results section.

The logistic regression analyses are still not clearly explained in section 2.3 (statistical analysis), although Table 3 does help illuminate the approach being used. In my understanding, each variable in the Variables/Category column was the primary predictor in the logistic regression models, and then the models were adjusted for the additional variables mentioned in the table footnote (Model 1: gender + school level, Model 2: gender + school level + grades + economic status + bullying + gaming disorder, etc). However, this doesn’t quite make sense to me, since you can’t adjust for gender in a model where gender is the primary predictor. In explaining the statistical analysis, it should be clear what the primary predictor was (gender, school level, etc), and what, if any, additional covariates were included in that particular model.

That said, the data provided in Tables 3 and 4 are very helpful for understanding what is being analyzed and what the findings are.

  • We revised the sentences a little bit to clarify the meaning of model 1,2, and 3 of table 3 and 4.

: The effect of each variable as the primary predictor on the risk of NSSI was evaluated in the regression models, mutually adjusted for gender, school level, economic status, bullying victim, gaming disorder, and dpressive symptom (Table 3). Stratifed analysis by gender was performed with same covariates (Table 4). 

English grammar continues to be a problem; while the text is not unreadable, spelling errors and grammatical mistakes detract from the clarity of the data and ideas presented.

  • We are going to request grammar editing to the Journal. Sorry about the confusion.

Minor Comments

The authors cite something called the OECD, but this is not defined. All acronyms should be fully spelled out with the first use and then abbreviated thereafter.

  • Revised as suggested

There still aren’t any specific hypotheses about what the authors thought they would find in the introduction. I realize this seems pedantic since the text mentions stratifying analyses by gender, but stratifying by gender only makes sense if you expect a gender difference (which you do, given the relevant suicide data), and so your specific hypothesis would be that you expected females to have higher rates of NSSI than males. The point in mentioning this, by the way, is to disclose to the reader your expectation bias in conducting the study – you fully expected that females would have higher rates of NSSI and now the reader can assess if your data really supports that expectation.

  • Added as suggested.

: Since the poor socioeconomic environment is also highly likely to have adolescents exposed to negative incidents, the risk of selfinjury according to family’s economic status needs to be evaluated. Little studies, however, reported the gender differences in the factors associated with NSSI in Korea.

Suicide is a significant problem for South Korean adolescents and is worse among females. Because the majority of adolescents’ lives exist either in school or at home, it is appropriate to evaluate NSSIs by sampling adolescents from across school environments. Therefore, this study was conducted to estimate the prevalence of NSSIs and to evaluate the related factors with NSSIs in multiple schools in a single city in Gyeonggi province, nearby Seoul Metropolitan city, and analyses were stratified for gender. 

I actually think it is appropriate to find that NSSI rates decrease significantly as students get older (i.e., from middle school to high school), as this follows observed data from around the world. NSSI peaks in adolescents and tapers off into adult hood except in seriously mentally ill populations, where self-harm becomes a staple of emotion regulation.

  • It was mentioned in the discussion part.
